# Investigation of Autonomic Dysfunction in Alzheimer’s Disease—A Computational Model-Based Approach

**DOI:** 10.3390/brainsci13091322

**Published:** 2023-09-14

**Authors:** Sajitha Somasundaran Nair, Mini Maniyelil Govindankutty, Minimol Balakrishnan, Krishna Prasad, Talakad N. Sathyaprabha, Kaviraja Udupa

**Affiliations:** 1Model Engineering College, Cochin University of Science and Technology, Kochi 682022, India; 2Department of Neurophysiology, National Institute of Mental Health and Neurosciences, Bengaluru 560029, India

**Keywords:** Alzheimer’s disease, autonomic dysfunction, heart rate variability, computational model, neurotransmitter kinetics

## Abstract

(1) Background and Objective: Alzheimer’s disease (AD) is commonly accompanied by autonomic dysfunction. Investigating autonomic dysfunction’s occurrence patterns and severity may aid in making a distinction between different dementia subtypes, as cardiac autonomic dysfunction and AD severity are correlated. Heart rate variability (HRV) allows for a non-invasive assessment of the autonomic nervous system (ANS). AD is characterized by cholinergic depletion. A computational model of ANS based on the kinetics of acetylcholine and norepinephrine is used to simulate HRV for various autonomic states. The model has the flexibility to suitably modulate the concentration of acetylcholine corresponding to different autonomic states. (2) Methods: Twenty clinically plausible AD patients are compared to 20 age- and gender-matched healthy controls using HRV measures. Statistical analysis is performed to identify the HRV parameters that vary significantly in AD. By modulating the acetylcholine concentration in a controlled manner, different autonomic states of Alzheimer’s disease are simulated using the ANS model. (3) Results: In patients with AD, there is a significant decrease in vagal activity, sympathovagal imbalance with a dominant sympathetic activity, and change in the time domain, frequency domain, and nonlinear HRV characteristics. Simulated HRV features corresponding to 10 progressive states of AD are presented. (4) Conclusions: There is a significant difference in the HRV features during AD. As cholinergic depletion and autonomic dysfunction have a common neurological basis, autonomic function assessment can help in diagnosis and assessment of AD. Quantitative models may help in better comprehending the pathophysiology of the disease and assessment of its progress.

## 1. Introduction

Alzheimer’s disease (AD) is a global neurodegenerative condition and a common cause of dementia [1]. Autonomic dysfunction accompanies AD. The control of the cardiovascular and autonomic processes is largely dependent on the cholinergic system and its neurotransmitter Acetylcholine (*ACh*). Right from the pre-clinical stage, AD notably affects cholinergic system. Acetylcholine is a major neurotransmitter of parasympathetic nervous system. In AD, the synthesis of *ACh* is hindered. It is observed that the choline acetyltransferase activity is lowered to 35% to 50% of normal levels in AD. This enzyme catalyzes the production of *ACh* from choline and Acetyl coenzyme A. Additionally, synaptic reuptake of choline, which is necessary for replenishing the primary acetylcholine store for succeeding rounds of neurotransmission, is decreased to 60% of normal levels in AD. According to direct testing, *ACh* synthesis levels are lowered by 50% in affected patients [2].

It has been shown that in AD patients, increasing dementia severity is significantly linked with declining levels of *ACh* production [3]. The repercussions of Alzheimer’s disease extend beyond the individual, causing financial hardship for families due to the cost of care. Early detection of AD can yield numerous benefits, improving health, well-being, and financial stability for affected individuals and their families. Adopting a proactive approach will benefit both those impacted and society as a whole [4].

Investigation of autonomic functions in AD has clinical significance. It helps us to understand the extent of autonomic failure and hence precautions can be taken against postural dizziness, syncope, and falls [5]. As autonomic dysfunction is a novel biomarker of AD, it is important to monitor the changes in autonomic dysfunction for diagnosis and assessment of the progress of the disease. Moreover, it is useful to conduct autonomic function assessment for elderly demented patients to prevent morbidities related to abnormal sympatho vagal functions [5,6].

In the US, AD is ranked as the sixth leading cause of death, impacting around 6.5 million Americans aged 65 and older in 2022 [7]. The projected statistics indicate a significant increase in AD cases, with an estimated 13.8 million affected individuals in the US by 2050, and a new case occurring every 33 s. India faces a similar challenge, as the aging population is expected to reach 19.1% of the total by 2050. Consequently, dementia prevalence is predicted to rise, affecting around 11.4 million people in India by 2050 [8].

In AD patients, due to the significant loss of cortical perivascular cholinergic nerve terminals, there is a reduction in cerebral blood flow. The cholinergic-vascular hypothesis was developed as a result of the investigation of this correlation between cerebral blood flow and cholinergic inactivity [9,10]. In AD, various brain regions associated with cholinergic functions are affected, including the hypothalamus, locus coeruleus, cerebral neocortex, insular cortex, and brain stem. These structures have a crucial role in the functions of the ANS also. Therefore, cholinergic function impairment may result in autonomic dysfunction [11]. Autonomic dysfunction in general and parasympathetic dysfunction, in particular, are reported in mild cognitive impairment [12]. This may be due to early neuroanatomical and neurochemical changes in the central autonomic network in AD.

Neelesh Gupta et al. have presented a study of autonomic function tests in a group of AD patients and compared the outcomes with age and gender-matched healthy volunteers [2]. Low-frequency power in normalized units (LF nu) and the LF/HF ratio (sympathovagal ratio) were both significantly higher (*p* < 0.05) in the Alzheimer’s group, while high-frequency power in normalized units (HF nu) was considerably low (*p* < 0.05). The LF/HF ratio and disease severity are positively correlated. They concluded that parasympathetic activity is significantly reduced in AD patients. Additionally, they noticed that patients had sympathetic dominance and sympathovagal imbalance. This might be attributable to cholinergic depletion-induced central autonomic dysfunction. In addition, the severity of the disease grew in direct proportion to autonomic dysfunction.

The paraventricular nucleus, dorsomedial nucleus, lateral hypothalamic area, posterior hypothalamic nucleus, and mammillary nucleus are some of the central nervous system structures that are related to the sympathovagal network. Both the sympathetic and parasympathetic outflow is controlled by these regions. The rostral and ventrolateral caudal medulla, raphe nuclei, and locus coeroleus are the extra-hypothalamic structures associated with the sympathetic drive. The amygdala, dorsal motor nucleus of the vagus, and nucleus ambiguous are the other extra-hypothalamic structures that regulate the parasympathetic drive [13]. It is seen that about 50% loss of neurons takes place in the amygdala during the AD state, and there is severe volume reduction [14]. The hypothalamus, locus coeruleus, and insular cortex are a few of these areas that are mainly implicated in AD. As a result, it makes sense to assume that sympathovagal drive and the pathophysiology of AD are related [15].

According to Braak et al., autonomic dysfunction may exist prior to the start of dementia’s clinical symptoms. Thus, autonomic dysfunction is a strong independent predictor of degeneration of the nervous system [11]. Acetylcholinesterase activity levels in blood and cardiac autonomic dysfunction as determined by HRV evaluation are significantly correlated in AD patients. This demonstrates that a cholinergic deficit in the peripheral autonomic nervous system (ANS) may be the cause of autonomic cardiac dysfunction in AD patients [16]. Dias et al. have demonstrated that cholinesterase inhibitors can modify autonomic function in patients with AD. Treatment with these medications did, in fact, lead to an improvement in the behavior of the ANS [17]. Kajsa Stubendorff et al. have conducted a study on the influence of autonomic dysfunction on the survival of patients with dementia. It has been reported that autonomic function assessment has clinical importance not only to avoid accidents and falls but also as a predictor of survival [18].

Some of the autonomic function tests are Heart Rate Variability (HRV) analysis, Ewing’s battery tests, Baroreflex function tests, and plasma epinephrine and norepinephrine (NE) measurements. HRV analysis test involves the acquisition of ECG from the patient and analysis of this signal in the time, frequency, and nonlinear domain. This is a noninvasive procedure, is not an expensive method, and is quite easy to perform and repeat as required. Routine HRV analysis in AD patients may help as an indirect way of assessing the severity of the disease. Moreover, this analysis will help us to predict the complications that may arise in the future.

A perusal of the literature shows that there is a massive expansion of research in the field of AD in the last two decades. Scientists viewed it from biochemical, molecular, genetic, and epidemiological angles, thus augmenting our knowledge of AD. However, there is a large terrain yet to be explored about the underlying mechanism of the disease and its complexities. Cholinergic depletion leads to autonomic dysfunction in AD but the exact mechanisms by which it happens and the way in which it progresses are not known yet.

Mathematical models can be of use in this scenario since such models give us quantitative insight into the associated physiological mechanism. Models can help us in the design of targeted diagnosis. Also, drugs targeted at specific locations can be designed. Researchers have attempted to model HRV but most of them are utilizing nonlinear oscillators to represent the components of the ANS, and hence, cannot account for the physiological underpinnings. Physiology-based models seen in the literature are modeled at the cell level and have not simulated or validated the HRV.

The authors have developed a mathematical model of the ANS, based on physiology, which can simulate HRV [19]. The present study is about how this model can be used to simulate various autonomic states of the progress of AD. This approach might be useful in predicting the progress of the disease. ANS is responsible for heart rate regulation. Heart rate is suitably controlled by controlling the kinetics of acetylcholine and norepinephrine secreted from the parasympathetic and sympathetic nervous system, respectively. The developed ANS model explicitly models the neurotransmitter kinetics.

The sympathetic and parasympathetic nervous systems are dynamically balanced. When this equilibrium is maintained, the variability within the inter-beat interval is at its optimum level. This is necessary for the adaptability and flexibility of the cardiovascular regulatory system, which reflects a person’s physical and mental health. A tilt in the balance results in reduced HRV, which is a strong regressor of health issues. Patients with autonomic dysfunction, such as those with anxiety, depression, neurodegenerative illnesses, asthma, and sudden newborn death, are shown to have low HRV [20]. We applied the model to simulate HRV in both pathophysiologic and normal conditions.

The developed ANS model deals with neurotransmitter kinetics in detail. Hence, it is possible to represent various stages of cholinergic depletion and hence various degrees of autonomic dysfunction. Not just one, but a few parameters are available to simulate the cholinergic depletion state in the present mathematical model. This enhances the flexibility of the model.

## 2. Methods

### 2.1. Statistical Analysis of AD HRV Data

HRV values were utilized to conduct a comparison between AD patients and age- and gender-matched control subjects. The primary objective was to pinpoint the specific HRV parameters that exhibit changes in the context of Alzheimer’s disease (AD). Subjects were chosen from the Autonomic Function Testing (AFT) Laboratory database of the National Institute of Mental Health and Neurosciences (NIMHANS), Bengaluru, where the patients were referred to the AFT Laboratory from Departments of Neurology and Geriatrics Psychiatry for routine autonomic function evaluation. Based on the patient’s clinical history, a thorough clinical examination, and the DSM-V criteria, the diagnosis of probable Alzheimer’s disease was made. The patients were graded based on the Clinical Dementia Rating (CDR). The study excluded patients who used medicines that had an impact on autonomic function. Those who satisfied the criteria for the study’s eligibility were recruited after receiving informed consent in writing from both the patient and the caregivers. Before being evaluated, study participants were instructed to forgo alcohol, nicotine, coffee, and tea for 24 hrs. Additionally, it was suggested that they have a light meal three hours before the test and empty their bladder and bowels.

Short-term HRV data analysis is conducted in this study, hence, the physiologic data and simulated data are of short duration, typically 5–10 min. KUBIOS HRV 3.2.0 analysis software is used for signal analysis [21]. SPSS V.22 was used to conduct a non-parametric statistical study of the normal and AD HRV features [22]. A non-parametric test was chosen since every parameter differed from the normal distribution. The purpose of the test was to assess whether the descriptive statistical measures (Median) differed significantly between the normal and the AD HRV parameters. Since the study is based on 20 numbers of sample data and not on the entire population, it is unlikely that the sample difference and the population difference are the same.

A *p*-value < 0.01 is employed in the statistical significance test. Therefore, the likelihood of incorrectly concluding that a difference exists when there is not one is less than 1%. With the assumption that there is no dependence between the normal and AD HRV signals, a Mann–Whitney *U* Test is chosen [23]. After the key features reflecting AD are identified by the test, we decided to simulate HRV for various stages of the progress of Alzheimer’s disease.

### 2.2. Computational Model of ANS for HRV

The model framework is based on neurotransmitter kinetics. Norepinephrine and acetylcholine are the major neurotransmitters involved in the heart rate regulation process. Norepinephrine is from the sympathetic side and acetylcholine is from the parasympathetic side, respectively. The kinetics of these neurotransmitters at the neuroeffector junction, extra junctional space, and extra cellular matrix are explicitly modeled. The concentration of acetyl choline at the neuro effector junction and extra junctional space constitutes the vagal signal. Similarly, the concentration of norepinephrine at the neuro effector junction and extra junctional space constitutes the sympathetic signal. A combination of sympathetic and parasympathetic signals is used as the autonomic neural signal which is responsible for heart rate regulation [19].
(1)$$m\left(t\right)=m_{0}+\left(S\left(t\right)+V\left(t\right)\right)\;.$$
*S*(*t*) and *V*(*t*) are the sympathetic and vagal contributions, respectively, to the autonomic neural signal, $m\left(t\right).$ $m_{0}$ is a constant that can vary with individuals, it corresponds to the intrinsic heart rate. All factors contributing to heart rate regulation other than autonomic influence are lumped into the parameter,$\;m_{0}$. In our simulations, $m_{0}$ is kept as 1; $m\left(t\right)$ is strictly positive.

Rate variations of the heart are largely decided by the ANS. However, the hormonal system, circadian rhythmicity, and hemodynamic variables also have some influence on heart rate regulation but to a lesser extent. If the heart is denervated (autonomic control absent), for example, in the case of a transplant heart, $m_{0}$ will be the only factor contributing to heart rate regulation. To generate heartbeat interval series, we have connected a heart model following the ANS model. Detailed description of HRV generation using the ANS model is available in [19].

In comparison to the skeletal nervous system, the ANS has lower frequency of stimulation [24]. In general, Sympathetic system may receive an input once in every (20–40) s, while parasympathetic inputs may get stimulation once in every (2–4) s. The latter is primarily mediated by respiratory activity. We have used input frequencies from 0.02 Hz to 0.4 Hz, for stimulation. A small quantity of Gaussian noise with a zero mean and 10 ms standard deviation is also artificially supplied at the output. Biological systems have some amount of natural random variability and that is represented by this noise [25].

The present mathematical model is based on the kinetics of neurotransmitters *NE* and *ACh*. Heart rate variability is regulated by these neurotransmitters. Autonomic function parameters obtained on analysis of HRV reflects the activity of these neurotransmitters. Since *ACh* kinetics is modeled explicitly in the model, we can represent a choline depletion state (AD pathophysiology) using this model. It is straight forward that the autonomic function parameters get altered in AD state.

It is reported that acetylcholine synthesis is significantly reduced in AD [3]. To simulate an autonomic state corresponding to AD, in our model we reduced the concentration of *ACh* to 70% of the original. Subsequently, the vagal signal is reduced.

This results in a modification of autonomic neural signal, and hence, heart rate variability. Equation (1) depicts sympathetic and parasympathetic signals when there is a balance between sympathetic and parasympathetic activities (normal). By controlling the concentration of acetylcholine, we can modulate the vagal signal and represent various autonomic states. This implies that various stages of the pathogenesis of AD can be represented using the model. It can be used to forecast the growth pattern of the disease and assess how well a drug will work if it is taken.

Modulating the *Ach* concentration can be conducted at various levels in the ANS model. At the basic level, the number of vesicles loaded with *ACh* can be reduced, the concentration of *ACh* at the neuroeffector junction and/or extra junctional space can be reduced, the degree of interaction between vagal and parasympathetic signal can be modified to create a situation of vagal suppression. At a gross level, the vagal contribution to the autonomic signal also can be reduced. In this study, the concentration of acetylcholine at the neuroeffector junction is modulated. The model’s adaptability enables precise adjustment of a variety of parameters to replicate the autonomic states. This allows for the representation of the true physiological state. We fine-tuned the concentration of acetylcholine from 100% to 55% in discrete steps, thus modifying the vagal signal. We could generate a series of 10 model simulations, representing 10 various states of progress of AD.

### 2.3. HRV Features Reflecting Autonomic Dysfunction

The Peak frequencies in the Very Low-Frequency band (VLF), Low-Frequency band (LF), and High-Frequency band (HF), Low-Frequency power in the normalized unit (LF(nu)), High-Frequency power in the normalized unit (HF(nu)), and the LF/HF ratio are the main features in the frequency domain. Welch’s periodogram approach and autoregressive (AR) modeling are two techniques that are supported by KUBIOS for spectrum estimation [21,26]. The total power is calculated by integrating the spectrum estimates over the entire frequency spectrum, and the band powers are calculated by integrating the estimations over the frequency band limits. We have not considered VLF features in this study, since our data recordings and simulations were of short-term (5–10 min) duration.

Nonlinear analysis features were extracted from the Poincare plot. The Poincare plot depicts the relationship between the *n*th R-R interval, (*RRn*), and (*n* + 1)th R-R interval, *RR_n_*_+1_. The shape of the plot has diagnostic significance. An ellipse is fitted into the data points (RR*_n_*, RR*_n_*_+1_) oriented along the line of identity (LOI; line where RR*_n_* = RR*_n_*_+1_). The width and length of the ellipse are denoted by SD1 and SD2, respectively [26]. SD1 is the standard deviation of the points perpendicular to the LOI which quantifies the short-term variability and SD2 is the standard deviation of the points along the LOI which quantifies the long-term variability.

The standard deviation of normal-to-normal N-N intervals (SDNN) and the square root of the mean squared differences of successive N-N intervals (RMSSD) are two main features in the time domain. SDNN captures both short- and long-term variability within the R-R interval series while RMSSD is a measure of short-term variability.

Short-term recordings (5–10 min) are sufficient to examine LF (0.04–0.15 Hz) and HF (0.15–0.40 Hz) activities. Vasomotor tone and barro reflex activities are reflected in the LF band while respiratory activities are reflected in the HF band. A 24 h recording of HRV and its analysis will reveal the very low frequency (VLF) (0.0033–0.04 Hz) and ultra-low frequency (ULF) (<0.003 Hz) events. Core temperature regulation, metabolism, and the renin–angiotensin system activities contribute to ULF frequencies. Physical exercise, thermoregulatory, renin–angiotensin, and endothelial influences on the heart contribute to VLF power. Thus, comprehensive HRV analysis can reveal very useful functional aspects of the ANS [27].

## 3. Results

The statistical analysis suggests that SDNN, RMSSD, LF(nu), HF(nu), LF/HF ratio, SD1, and SD2 of the Poincare plot are the key HRV parameters that differ significantly from normal in AD. With the exception of SD2/SD1 of the Poincare plot, all parameters considered in this study were significantly different at a *p*-value of 0.01. Table 1 provides a summary of the findings. These results emphasize the fact that AD is characterized by declined autonomic function. ANS innervates all vital organs which are not under voluntary control. Hence, the observed cardiac autonomic dysfunction provides a strong indication that all these organs may be affected in the disease state.

The decrease in RMSSD and SDNN suggests a decrease in the variation in the time interval between successive cardiac cycles. In the disease state, the heart tends to be monotonous losing its self-regulatory capacity. Both the short-term variability (RMSSD) and overall variability (SDNN) are significantly compromised. The HF power (HF (nu)),quantifying parasympathetic activities are reduced. A higher-than-normal LF/HF ratio indicates that sympathetic dominance has taken over and the sympathovagal balance has been lost. It may be noted that all the frequency domain parameters are significantly different in AD patients compared to the normal group. Power spectrum analysis is considered a suitable method to distinguish AD patients from the control group in [28] also. The use of continuous wavelet transforms and Fourier transform are illustrated in [29] for establishing autonomic dysfunction in dementia.

The Poincare plot changes from its typical comet shape to that of a rod or even to a cluster of points as evidenced by the drop in SD1 and SD2. Very Few points are aligned perpendicular to LOI depicting reduced short-term variability.

As the HRV features showing significant variation in AD have been identified using statistical analysis, we decided to simulate AD pathogenesis using the ANS model. Initially, the model is used to simulate HRV for two different autonomic states.

(i)Balanced sympathovagal activities (healthy state)(ii)Vagal suppression due to reduction in acetylcholine concentration (AD disease state).

Figure 1 depicts the parasympathetic signal and the sympathetic signal generated for balanced sympathetic and parasympathetic activity (healthy). The solid line shows the sympathetic signal, and the dotted lines represent the vagal signal. This autonomic state is depicted by Equation (1) and corresponding simulated HRV features are listed in Table 2 (autonomic state 1). Vagal signal slightly outweighs the sympathetic signal in Figure 1. Slight parasympathetic dominance is observed in humans in resting conditions and sympathovagal balance corresponds to a healthy state [30].

Figure 2 shows the sympathetic and vagal signal generated when the concentration of acetylcholine is reduced to 70% of the original value. Consequently, a reduction in the vagal signal is observed. This will cause a change in the autonomic neural signal and thus modifying the autonomic state and heart rate variability. The corresponding simulated HRV features are shown in Table 2 (autonomic state 7).

To test the reliability of the model we simulated HRV for 10 different autonomic states starting from a balanced sympathetic–parasympathetic activity (sympatho vagal balance—0.56) to severe autonomic imbalance (sympatho vagal balance—9.74). These simulations represent the progressive states of AD. These HRV data are simulated by decreasing the concentration of acetylcholine from 100% to 55% in the model in 10 discrete steps. Table 2 illustrates the HRV features of the simulated HRV.

All the HRV parameters which showed significant variation in the statistical study are presented in Table 2. A considerable decrease in SDNN and RMSSD are observed; this is quite obvious as the overall variability (SDNN) and short-term variability (RMSSD) are reduced in AD.

A gradual increase in LF (nu) from 35.7 to 90.68 is observed as we progress from the normal state to a disease state. Conversely, the HF (nu) reduces from 63.75 to 9.32. It is clear that a sympatho vagal imbalance is created as we move from normal to state to disease state. LF/HF ratio increases from 0.56 to 9.74 in our simulations.

As the disease progress, the width of the Poincare plot, SD1 (dispersion of points in the perpendicular to LOI) decreases. The short-term variabilities are parasympathetically mediated and are represented by SD1 while long-term variabilities are sympathetically mediated and are represented by SD2. This observation is justifiable as the parasympathetic activity reduction and sympathetic activity dominance accompany the progress of the disease. Not only the parasympathetic system, the sympathetic system also gets affected during AD. The sympathetic response to the stress of standing cannot be tolerated by AD patients [31]. Autonomic dysfunction concerning sympathetic and parasympathetic systems during AD has been reported in [32].

There is interaction between the sympathetic and parasympathetic systems. Literature shows that there is mutual excitation and mutual inhibition between the branches of the ANS [33]. The reduction in vagal activity due to cholinergic depletion might have caused inhibition of sympathetic activity also. We have taken care of the interaction between sympathetic and parasympathetic activities in our model. Frequency domain analysis of the AD physiologic HRV data reveals a reduction in actual power in both the LF (to a less extent) and HF bands (to a great extent). Hence total power decreases in AD disease state. There is sympathetic dominance in the AD state; most of the total power is concentrated in the (LF) band. This phenomenon is captured by the model also. It may be noted that substantial reduction in amplitude of the parasympathetic signal and slight reduction in sympathetic signal are observed in Figure 2 compared to Figure 1. The actual value of power as shown on the Y-axis of Figure 4 (AD state) is less that on Figure 3 (balanced sympathovagal state). The LF and HF powers expressed in Table 2 are in normalized units (nu). Thus, the model faithfully reproduces AD pathological state.

These observations imply that the regulatory capacity of the cardiovascular system decreases as AD progresses. The results emphasize the fact that autonomic dysfunction is accompanied by AD and it is reflected in the frequency domain, time domain, and nonlinear parameters of HRV. These findings imply that the ANS model is robust to replicate the various stages of AD autonomic states.

Each patient in the AD patient group we took into consideration has a varied disease severity. After closely evaluating them, we discovered that several of the simulated HRV elements, particularly frequency domain properties, resemble the HRV features for AD patients. This indicates the possibility of making personalized models while treatment strategies are planned. Table 3 lists a set of three AD physiological HRV frequency domain features as well as three simulated AD HRV frequency domain features. These are comparable.

The first autonomic state is associated with balanced sympathovagal activity. Autonomic states 2 and 3 relate to moderate and advanced stages of AD, respectively. Figure 3, Figure 4 and Figure 5 also show the HRV frequency spectrum for each of these autonomic states. The resemblance in values between physiological and simulated HRV features is evident within each individual autonomic state. Based on the frequency domain parameters, we can conclude that the severity of the disease worsens as we move from autonomic states 1 to 3.

Figure 3a,b are the power spectrum corresponding to balanced sympathovagal activities (normal) for simulation and physiologic HRV, respectively. The spectra have distinct peaks in the LF and HF bands corresponding to sympathetic and parasympathetic activities. There is significantly more power concentrated in the HF band compared to the LF band making a sympathovagal ratio less than one in both cases. The area enclosed under LF and HF bands estimate the LF and HF power respectively. Thus, the model simulated data and physiologic HRV data have similar frequency spectra for the autonomic state of sympathovagal balance (autonomic state 1).

Figure 4a,b are the power spectra of simulated HRV and physiologic HRV corresponding to the AD disease state. In both cases, significant power is concentrated in the low-frequency band, and LF peak frequency is also present. The power in the HF band has been reduced significantly; sympathovagal balance is lost. When Figure 3 and Figure 4 are compared, we can observe the decline in the high frequency content in Figure 4.

Figure 5a,b are the power spectra for an advanced state of AD. In Figure 5a—simulation and Figure 5b—AD physiologic HRV, the high-frequency peak has almost vanished, and very little power is contained in the high-frequency band. Almost the entire power is concentrated in the LF band; autonomic dysfunction is notable. A remarkable sympathetic dominance and severe sympathovagal imbalance are observed.

A careful examination of the results shown in Table 2 and Table 3 and Figure 3, Figure 4 and Figure 5 reveals that the quantitative ANS model is robust to reproduce various autonomic states of AD.

Alzheimer’s disease is a complex neurodegenerative disease. Many complexities develop as the disease progress, like cognitive decline, neuromuscular problems, behavioral changes, etc. which may further alter the autonomic states. These factors are not taken into consideration while simulating the disease states here. We have simulated the alterations in autonomic states by modulating the concentration of *ACh* as the disease progresses.

## 4. Discussion

Alzheimer’s disease is a complex neurodegenerative disorder that primarily affects cognitive functions, memory, and behavior. To effectively address the disease and develop targeted interventions, a deep grasp of its neurological underpinnings is imperative. We still have a limited understanding of the neurobiology, diagnosis, management strategies, and treatment options for AD. Quantitative models can be useful in this scenario. It helps in better understanding the neurochemical alterations; the accompanying physiological changes and the elements affecting the progress of the disease.

Autonomic dysfunction is an innovative biomarker of Alzheimer’s disease. Being a complex neurodegenerative disorder, diagnosis and estimation of the progress of the disease are difficult. There comes the significance of autonomic dysfunction assessment. Since cholinergic depletion invariably affects the autonomic function parameters, the diagnosis and assessment of the progress of the disease can be enhanced by analyzing autonomic function parameters.

Among various autonomic function assessment techniques, heart rate variability (HRV) analysis stands out due to its noninvasiveness and patient-friendly nature. HRV analysis, providing insights across time, frequency, and nonlinear domains, offers a comprehensive reflection of an individual’s autonomic function status.

The autonomic nervous system (ANS) influences numerous involuntary physiological processes. Evaluating ANS integrity becomes imperative for ensuring the optimal functioning of these organs. Beyond its cardiovascular insights, HRV parameters offer a unique perspective on the functional state of all ANS-innervated organs. Thus, the HRV signal acts as a window through which we can observe the broader impact of ANS, encompassing various systems, including the cardiovascular system.

This study explores the use of a computational ANS model to comprehend the neurobiological underpinnings of autonomic dysfunction in Alzheimer’s disease (AD). The statistical analysis highlights significant differences in HRV features between the AD and normal states. The statistical analysis reveals that the HRV features, namely, the LF power, HF power and LF/HF ratio (frequency domain), SDNN and RMSSD (time domain), and SD1 and SD2 (nonlinear domain) show significant differences in AD state compared to the normal state.

The LF power signifies sympathetic activity in the HRV signal, while the HF power reflects parasympathetically mediated activity. The LF/HF ratio indicates the balance between the sympathetic and parasympathetic systems. A slightly dominant parasympathetic state corresponds to health, with an ideal LF/HF ratio being <1.

In the context of AD, cholinergic depletion leads to decreased acetylcholine (*ACh*) concentration, resulting in diminished vagal signals and HF power. This disrupts the sympathetic-parasympathetic balance, causing an increase in the LF/HF ratio. Disease progression leads to declining HF power and an increasing LF/HF ratio, as detailed in Table 2. Data in the table illustrates normalized power, revealing that a larger portion of total power resides in the LF band as the disease advances. This shift does not imply increased actual sympathetic power but rather a decrease. Both short-term and overall variabilities decrease. Most of the power is concentrated in the LF band, signaling sympathetic dominance in contrast to the normal state.

Diminished parasympathetic activity raises the average heart rate, resulting in reduced R-R interval variability (SDNN and RMSSD). These observations indicate compromised ANS integrity and reduced cardiovascular regulatory capacity.

Diagnosis of AD usually happens late because behavioral changes, cognition decline, etc. occur in later stages. As autonomic function decline is observed in the early onset of the disease, autonomic function tests will increase confidence in diagnosing AD. Early diagnosis of Alzheimer’s disease helps in timely intervention and treatment for effective disease management and progression mitigation.

Autonomic function testing may also be used in dementia subtyping. The severity of AD and autonomic dysfunction are positively correlated. Hence investigation of autonomic functions can be used to assess the severity of the disease. However, AD is not the only cause of autonomic dysfunction.

Table 2 illustrates ten progressive AD states simulated using the ANS model. A comparison of three HRV simulations with three physiologic HRV features, in nearly identical autonomic states, is provided. The HRV features of AD physiologic HRV data and simulated HRV data are comparable. This observation is emphasized by the frequency spectrum representations in Figure 3, Figure 4 and Figure 5. The findings, summarized in Table 2 and Table 3 and Figure 3, Figure 4 and Figure 5, serve to validate our model simulations.

The high-frequency content significantly reduces in the disease state, and it gets worsened as the disease advances. Given that cholinergic activity is essential for the parasympathetic system’s operation and that parasympathetic activities are represented in the high-frequency band in the HRV frequency spectrum, this outcome is to be expected. Progressive cholinergic depletion is reflected in the HRV frequency spectrum as diminished high-frequency power as depicted in Figure 3, Figure 4 and Figure 5.

The gradual increase in the LF/HF ratio, shifting from low to high values, aligns with expectations as the sympathovagal balance is disrupted, leading to sympathetic predominance. This trend is evident in both physiologic and simulated HRV analyses. Table 2 and Table 3, along with Figure 3, Figure 4 and Figure 5, quantitatively capture these physiological shifts.

Observations of declining SDNN and RMSSD values, seen in both physiologic and simulated HRV analyses, indicate reduced variability among R-R interval sequences as the disease progresses. This suggests diminishing cardiovascular adaptability and self-regulation due to neurodegeneration, potentially rendering the heart more vulnerable to disorders. The decline in SD1 value is notable, linked to reduced parasympathetic activity and less dispersion of points perpendicular to the line of identity (LOI). This decrease in overall variability is accompanied by a rise in the SD2/SD1 ratio, signifying an imbalanced operation of the ANS branches with pronounced sympathetic dominance.

The current study indicates the possible use of personalized models for AD management. It empowers clinicians to develop individualized treatment plans, leading to better patient outcomes and improved management of this complex neurodegenerative condition. Many brain centers are affected in AD state, these brain centers are involved in multiple functions. Hence, the symptoms exhibited by all the patients need not be the same. Moreover, comorbidities are also to be considered. Hence a personalized approach will be more suitable in the management of such neurodegenerative states.

Also, models can predict the kind of improvements the medicine can bring about. The current autonomic function status is revealed by the HRV parameters. Simulations can provide the clinician with an idea regarding the possible changes in the disease state likely to occur, the drugs to be optimized and the precautions to be taken against accidents. Personalized models might help in predicting future complexities during the course of the progress of the disease.

## 5. Conclusions

AD is a chronic, slowly progressing, neurodegenerative disease. Although an appreciable amount of research is found in literature about disease pathophysiology and management strategies, our understanding of the disease is still limited. The complexity of Alzheimer’s disease necessitates a multi-disciplinary approach to research. Such collaboration brings together experts from medicine, science, and engineering. Combining expertise from neuroscience and technology can lead to the development of more accurate and sensitive diagnostic tools, facilitating early detection and intervention. There is a large space for improvement in terms of developing reliable biomarkers, accurate diagnostic tools, and targeted medicines. Mathematical models may be useful to get an insight into the neurophysiological alterations accompanying the disease, identify novel biomarkers, and design effective diagnostic tools.

The study makes use of a computational model of the ANS, developed by the authors, to simulate various autonomic states of AD. HRV parameters reflecting AD pathophysiology are identified using a statistical analysis of AD HRV data. Cholinergic depletion is reported in AD. Moreover, the decline in autonomic function parameters and disease severity is positively correlated. Progressive states of AD are simulated using the model by carefully regulating the concentration of *ACh*. The simulated HRV features for AD exhibit a remarkable resemblance to the corresponding physiological HRV features observed in individuals with AD. Mathematical models can help to identify the possible target of medicine, assess the progress of disease and evaluate drugs. The model frame work can be expanded to develop personalized models. Personalized medicine and models in Alzheimer’s disease offer the potential for more precise and effective diagnosis, treatment selection, risk prediction, and monitoring of disease progression.

Quantitative models can be utilized as an adjunct to present AD diagnosis and management strategies. Mathematical models offer a promising avenue for enhancing Alzheimer’s disease management by tailoring interventions to individual patient characteristics, optimizing treatment efficacy, and ultimately improving patient outcomes.

## Figures and Tables

**Figure 1 brainsci-13-01322-f001:**
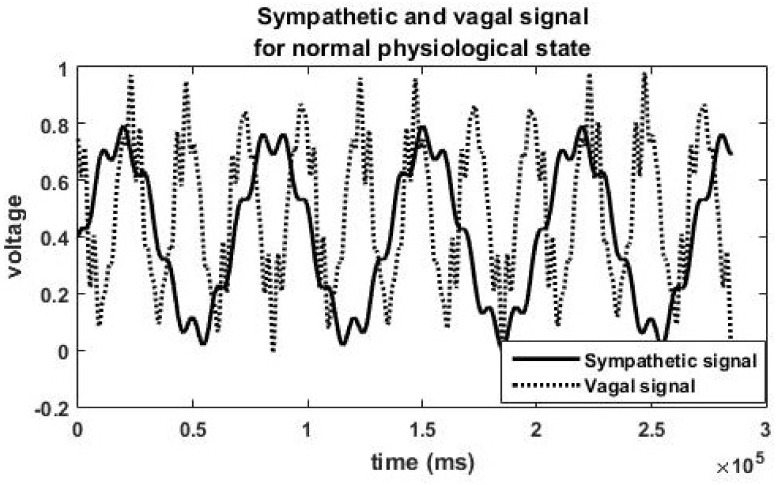
Simulated sympathetic signal and parasympathetic signal for a healthy normal person with balanced sympathovagal activities.

**Figure 2 brainsci-13-01322-f002:**
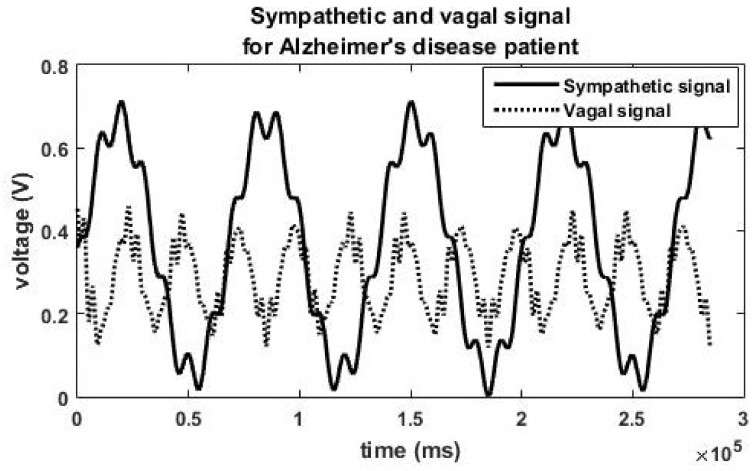
Simulated sympathetic signal and parasympathetic signal for an Alzheimer’s disease patient having Vagal inhibition and sympathetic dominance.

**Figure 3 brainsci-13-01322-f003:**
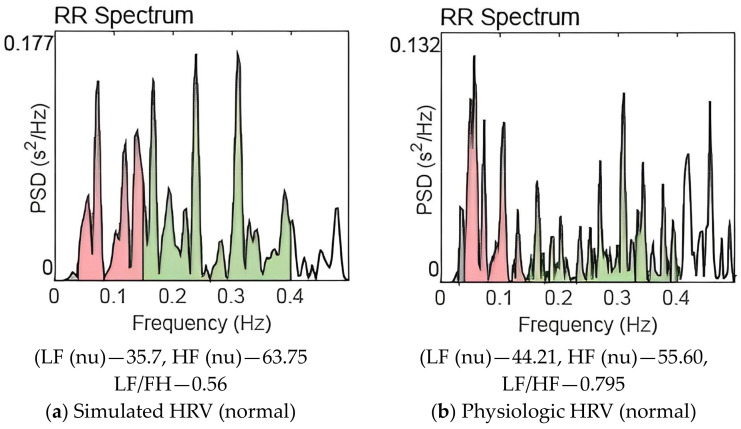
Power spectrum of HRV for balanced sympathovagal activities. (LF (0.04–0.15 Hz), HF (0.15–0.40 Hz)).

**Figure 4 brainsci-13-01322-f004:**
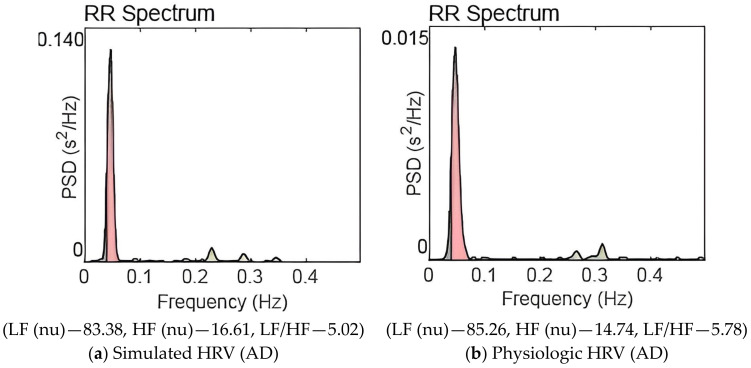
Power spectrum of HRV for AD pathogenesis. (LF (0.04–0.15 Hz), HF (0.15–0.40 Hz)).

**Figure 5 brainsci-13-01322-f005:**
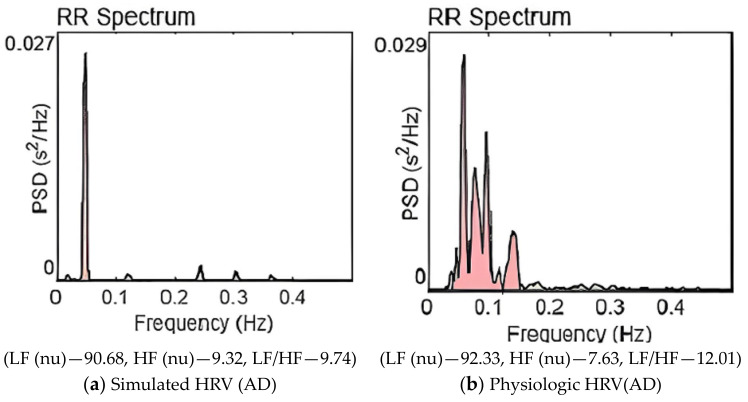
Power spectrum of HRV for the advanced state of AD pathogenesis. (LF (0.04–0.15 Hz), HF (0.15–0.40 Hz)).

**Table 1 brainsci-13-01322-t001:** Statistical test results for comparison of HRV features between Alzheimer’s disease patients and healthy volunteers.

HRV Parameters	Normal	Alzheimer’s Disease	Mann–Whitney U Value	*p* Value
SDNN	36.3	10.9	1	* <0.001
RMSSD	41.4	10.4	0	* <0.001
LF(nu)	37.32	69.58	14	* <0.001
HF(nu)	62.12	29.95	14	* <0.001
LF/HF	0.601	2.323	14	* <0.001
SD1	29.3	7.4	0	* <0.001
SD2	37.8	12.3	6	* <0.001
SD2/SD1	1.401	1.68	87.5	0.048

HRV: Heart Rate Variability, SDNN: Standard deviation of the normal R-R intervals, RMSSD: square root of the mean squared differences of successive NN intervals, LF(nu): Low-frequency power expressed in normalized units, HF(nu): High-frequency power expressed in normalized units, SD1 and SD2: measures of the Poincare plot, Values expressed as median, * *p* < 0.01.

**Table 2 brainsci-13-01322-t002:** HRV features for progressing AD states are simulated using the ANS model.

Autonomic State	LF Power (nu)	HF Power (nu)	LF/HF	SD1	SD2	SD2/SD1	SDNN	RMSSD
1	35.70	63.75	0.56	127	130.6	1.02	128.8	180
2	57	42	1.37	59.7	95.5	1.60	79.5	84.1
3	76.35	23.62	3.64	32	75.1	2.35	57.6	45.2
4	83.38	16.61	5.02	18.1	57.4	3.17	42.5	25.5
5	86.34	13.68	6.30	13.6	47.4	3.41	34.8	19.2
6	87.80	12.14	7.23	11	39.7	3.60	29.1	15.4
7	88.726	11.27	7.80	8.9	33.2	3.71	24.3	12.6
8	89.33	10.66	8.37	7.3	27.7	3.77	20.3	10.3
9	89.90	10	8.92	6	23	3.85	16.9	8.4
10	90.68	9.32	9.74	4.7	18.6	3.95	13.6	6.7

**Table 3 brainsci-13-01322-t003:** HRV features of Alzheimer’s disease patients. (Physiologic HRV and simulated HRV).

Autonomic State	HRV Signal	LF Power (nu)	HF Power (nu)	LF/ HF
1	simulation	35.70	63.75	0.56
	Normal Physiologic HRV	44.21	55.60	0.795
2	simulation	83.38	16.61	5.02
	AD Physiologic HRV	85.26	14.74	5.78
3	simulation	90.68	9.32	9.74
	AD Physiologic HRV	92.33	7.63	12.01

## Data Availability

The data sets used during the current study are available from the corresponding author upon reasonable request.

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
