# Peer review of "Investigation of Autonomic Dysfunction in Alzheimer’s Disease—A Computational Model-Based Approach"

_brainsci, 2023, doi:10.3390/brainsci13091322_

Round 1
Reviewer 1 Report
Comments and Suggestions for Authors
The authors have presented a manuscript titled "Investigation of Autonomic Dysfunction in Alzheimer's Disease: A Computational Model-Based Approach." The primary objective of this paper is to comprehensively examine autonomic dysfunction in Alzheimer's disease (AD) through the analysis of heart rate variability (HRV) measures and the utilization of a computational model of the autonomic nervous system (ANS). The study aims to identify significant variations in HRV parameters between AD patients and healthy controls, specifically evaluating vagal activity, sympathovagal imbalance, and changes in the time domain and nonlinear HRV characteristics. Furthermore, the paper seeks to simulate HRV for different autonomic states in AD using the ANS model, enabling valuable insights into the autonomic dysfunction associated with the disease. Ultimately, the goal is to underscore the potential of HRV analysis and computational modeling in assessing autonomic function, thereby aiding in diagnosing and evaluating AD while advancing our understanding of its pathophysiology and progression. The manuscript is interesting and fits well with the scope of the Journal.
Here are my specific comments:
- The introduction provides valuable information about the significance of AD, its prevalence, and the need for early detection. It also highlights the potential of mathematical modeling and autonomic function assessment to understand the underlying mechanisms and predict AD progression. However, there are a few suggestions for improvement. The introduction starts with statistics and prevalence data, which can be condensed to provide a concise overview. Consider focusing on key points and transition to the main topic of autonomic dysfunction in AD.
- The introduction could be organized more clearly by first establishing the link between AD and autonomic dysfunction and then introducing the importance of autonomic function assessment in understanding and managing the disease.
- While the introduction mentions the expansion of research on AD, it could explicitly highlight the need for further investigation into the underlying mechanisms of autonomic dysfunction and the potential role of mathematical modeling in addressing this gap.
- Some sentences are long and complex, making the text difficult to follow. Simplify the sentence structures and divide lengthy paragraphs into smaller ones to improve readability.
- Claiming the novelty and motivation for the manuscript at the end of the introduction is a good practice to highlight the unique contribution and significance of the study.
- Methods are described in detail.
- Results are generally well-presented, but Figures 1 and 2 have poor visibility. Their resolution should be improved.
- Discussion should be improved. Instead of simply stating that autonomic dysfunction is an innovative biomarker of AD, please briefly explain why it is considered innovative and how it contributes to understanding the disease.
- Explain briefly the significance of HRV analysis as a comprehensive assessment of the cardiovascular system's health and its role in reflecting the integrity of organs innervated by the autonomic nervous system.
- Connect the observed changes in HRV parameters (such as the reduction in high-frequency power, increase in LF/HF ratio, and decline in SDNN, RMSSD, SD1) to the underlying cholinergic depletion and sympathetic dominance in AD. Explain how these changes reflect the deteriorating integrity and adaptability of the cardiovascular system.
- Emphasize the importance of autonomic function testing in identifying AD at an early stage, which can aid in disease management, slowing down its progression, and preventing complications such as falls and accidents.
12. Here are some specific suggestions for improving the conclusion: Highlight that despite significant advancements in understanding AD, there is still much to uncover regarding its neurobiology, diagnosis, management strategies, and treatment options. It will emphasize the ongoing need for exploration and investigation.
- Clearly state the specific contributions of your study in addressing the research gap. For example, mention how utilizing a computational model of the ANS provided insights into the neurobiological basis of autonomic dysfunction observed in AD.
- Stress the value of personalized medicine and discuss how personalized mathematical models have the potential to revolutionize AD management by enabling tailored treatment approaches. Elaborate on how these models can provide insights into individualized intervention responses, thereby optimizing patient care and outcomes.
- Conclude with a strong statement.
Comments on the Quality of English Language
Minor editing is required.
Author Response
Dear sir,
Thank you for giving me insightful comments and suggestions on my manuscript titled " Investigation of Autonomic dysfunction in Alzheimer’s disease -A computational model based approach".
We have done our level best to answer your queries and incorporate your suggestions.
Kindly go through the attached files to see response to all the suggestions and comments you have made. If you have any more comments/queries/suggestions please let us know, we are happy to make modifications as per your advice.
Thank you very much for your time and efforts
Sajitha S (Corresponding author)

Reviewer 2 Report
Comments and Suggestions for Authors
Autonomic dysfunction usually coexists with Alzheimer's disease (AD). Given that cardiac autonomic dysfunction and the severity of AD are associated, examining the frequency and severity of autonomic dysfunction may help distinguish between various dementia subtypes. The autonomic nervous system (ANS) can be evaluated non-invasively using heart rate variability (HRV). The cholinergic system is depleted in AD. To simulate HRV for different autonomic states, an ANS computational model based on acetylcholine and norepinephrine kinetics is used. The model is adaptable enough to alter the acetylcholine concentration in accordance with various autonomic conditions. Using HRV measurements, 20 clinically plausible AD patients are contrasted with 20 age- and gender-matched healthy controls. The HRV parameters that differ significantly in AD are found using statistical analysis. The ANS model is used to simulate various autonomic states of Alzheimer's disease by carefully adjusting the acetylcholine concentration. A considerable decline in vagal activity, a sympathovagal imbalance with a predominance of sympathetic activity, a change in the temporal domain, and nonlinear HRV features are all present in AD patients. Ten successive phases of AD are represented by simulated HRV characteristics. The HRV characteristics during AD differ considerably. Autonomic function testing can aid in the diagnosis and evaluation of AD since cholinergic depletion and autonomic impairment have a same neurological substrate. In order to better understand the pathophysiology of the disease and evaluate its severity, quantitative models maybe used.
- It is not clear whether the simulated data is used or the real data was also incorporated to check the validity of the model.
- The model seems simple. Whether the brain’s data is simply being modeled? Please clarify.
- Whether the model can be applicable to majority of the patients or it’s a person’s specific?
- This model do incorporate the patients age or other demographics? This should be clear in the methods section.
- The discussion is completely not available. How the authors claimed that, the model is comparable to the existing results? No such comparison is provided.
- The reference section is too weak. The authors need to add the latest and old studies for better comparison of the results.
- The reference section is poorly managed. The authors need to double check the correctness and format.
Author Response
Dear sir,
Thank you for your insightful comments and suggestions on our manuscript titled "Investigation of Autonomic dysfunction in Alzheimer’s disease- A Computational Model based approach".
We have done our level best to answer your queries and incorporate your suggestions. If you have any more suggestions/queries please let us know, we are happy to make modifications as required.
Kindly see the attached file for responses to your comments. All these changes have been effected in the revised manuscript also.
Thank you for your time and efforts
Sajitha S (Corresponding author)

Round 2
Reviewer 2 Report
Comments and Suggestions for Authors
The conclusion section is too large. It should be shortened. Such long conclusion is not applicable.
Author Response
Dear sir,
I hope this email finds you well. We sincerely appreciate your time and effort in reviewing our manuscript titled "Investigation of Autonomic dysfunction in Alzheimer’s disease - A Computational model based approach" Your feedback and insights have proven invaluable in improving the quality and rigor of our work. We have carefully considered your comments and suggestions, and I am pleased to present our responses and the revised version of the manuscript.
Please refer to the attached file for a comprehensive overview of the changes made.
We genuinely value your expertise and insights. If you have any further suggestions or concerns, please do not hesitate to let us know. Your feedback is crucial in ensuring the highest quality of our work.
Thank you once again for your time and attention to our manuscript. Hope the revised version of the manuscript better meets your expectations.
Thank you
Sajitha S (Corresponding author)
